# Identification and Validation of Novel Potential Pathogenesis and Biomarkers to Predict the Neurological Outcome after Cardiac Arrest

**DOI:** 10.3390/brainsci12070928

**Published:** 2022-07-15

**Authors:** Qiang Zhang, Chenyu Zhang, Cong Liu, Haohong Zhan, Bo Li, Yuanzhen Lu, Hongyan Wei, Jingge Cheng, Shuhao Li, Chuyue Wang, Chunlin Hu, Xiaoxing Liao

**Affiliations:** 1Department of Emergency Medicine, The Seventh Affiliated Hospital, Sun Yat-sen University, Shenzhen 518107, China; zhangq555@mail2.sysu.edu.cn (Q.Z.); liuc65@mail2.sysu.edu.cn (C.L.); Libo1@sysush.com (B.L.); luyzh6@mail.sysu.edu.cn (Y.L.); chengjg5@mail2.sysu.edu.cn (J.C.); Wangchy99@mail2.sysu.edu.cn (C.W.); 2Department of Emergency Medicine, The First Affiliated Hospital, Sun Yat-sen University, Guangzhou 510080, China; zhangchy73@mail2.sysu.edu.cn (C.Z.); zhanhh6@mail2.sysu.edu.cn (H.Z.); weihy9@mail.sysu.edu.cn (H.W.); laish3@mail2.sysu.edu.cn (S.L.)

**Keywords:** cardiac arrest, biomarkers, neurological prognosis, hub genes, weighted gene co-expression network analysis

## Abstract

Predicting neurological outcomes after cardiac arrest remains a major issue. This study aimed to identify novel biomarkers capable of predicting neurological prognosis after cardiac arrest. Expression profiles of GSE29540 and GSE92696 were downloaded from the Gene Expression Omnibus (GEO) database to obtain differentially expressed genes (DEGs) between high and low brain performance category (CPC) scoring subgroups. Weighted gene co-expression network analysis (WGCNA) was used to screen key gene modules and crossover genes in these datasets. The protein-protein interaction (PPI) network of crossover genes was constructed from the STRING database. Based on the PPI network, the most important hub genes were identified by the cytoHubba plugin of Cytoscape software. Eight hub genes (RPL27, EEF1B2, PFDN5, RBX1, PSMD14, HINT1, SNRPD2, and RPL26) were finally screened and validated, which were downregulated in the group with poor neurological prognosis. In addition, GSEA identified critical pathways associated with these genes. Finally, a Pearson correlation analysis showed that the mRNA expression of hub genes EEF1B2, PSMD14, RPFDN5, RBX1, and SNRPD2 were significantly and positively correlated with NDS scores in rats. Our work could provide comprehensive insights into understanding pathogenesis and potential new biomarkers for predicting neurological outcomes after cardiac arrest.

## 1. Introduction

Cardiac arrest (CA) remains a major health burden around the globe, most often occurring in the community with a worldwide incidence of approximately 96 cases per 100,000 people per year [1]. More people are surviving cardiac arrest, and the overall results have improved through better emergency care, including early cardiopulmonary resuscitation (CPR), timely transport, and the implementation of a post-resuscitation care bundle [2,3]. Post-cardiac arrest brain injury is still a substantial cause of morbidity and mortality. For both shockable and nonshockable rhythms, targeted temperature management (TTM) and avoiding exposure to severe hyperoxemia are generally considered standard care after CA, because they improve neurological outcomes [4]. However, recent systematic reviews showed that TTM benefits on the neurological outcome are uncertain [5,6]. More new insights and therapeutic avenues after cardiac arrest are urgently needed. Many researchers have focused on continuous evaluation and prediction of neurological outcomes in comatose cardiac arrest survivors. Patients recovering from spontaneous circulation (ROSC) are comatose, two-thirds of them will die after CA. The majority of these deaths result from withdrawal of life-sustaining treatment (WLST) due to a predicted poor neurological outcome [7,8]. Therefore, neurological prognosis prediction is very important for clinicians when making appropriate treatment decisions and consulting with families about stopping life-sustaining treatment [9,10]. Various prognostic tools combined with the clinical examination are recommended to improve the accuracy of prognostic prediction in such patients [11,12]. Neurological examination is still the first step. Other tools include electroencephalogram (EEG), electrophysiological examination, brain imaging, and serological markers.

Among all the serum markers used to assess the neurological prognosis of comatose survivors, neuron-specific enolase (NSE) is the most widely studied measure for predicting neurological outcomes after CA. Research has shown the importance of NSE in outcome prediction, and it has been recommended for use in clinical practice [13,14]. In addition, the use of S100B and procalcitonin, either alone or in combination, for a neurological prognosis has gradually begun to be investigated, and these studies have yielded promising results, significantly increasing the value of serum markers for predicting neurological prognosis [13,15,16]. However, the discriminatory power of these tests is not ideal in individual patients, so efforts are still needed to find new biomarkers to optimize the management of individuals. New biomarkers would have multiple advantages. First, they must be beneficial to improve risk stratification, allowing for adjusting healthcare based on test results, optimizing the treatment of patients likely to survive and avoiding overmedication and the waste of healthcare resources. In addition, they need to provide information about the results to relatives at an early stage, limiting uncertainty and suffering.

The use of high-throughput sequencing/gene chips to mine new biomarkers is expected to meet these new stringent requirements and could elucidate the molecular mechanisms underlying poor prognosis. In a previous study, Iwahana et al. used RNA-sequencing to analyze genetic markers that can predict LV reverse remodeling in advanced non-ischemic heart failure and showed that the novel myocardial markers GADD45G and NDUFS5 can play this predictive role [17]. In addition, Chen et al. used RNA-seq analysis to identify genes associated with cognitive impairment after cardiac arrest and cardiopulmonary resuscitation [18]. Gene expression profiling has the capability to identify biomarkers associated with response to brain injury [19]. However, no studies explore the serum genetic markers’ relationship with prognosis after cardiac arrest. Therefore, this study was performed.

Herein, we identified differentially expressed genes (DEGs) between the high and low cerebral performance category (CPC) score groups by analyzing the mRNA expression profiles and clinical information of GSE29540 and GSE92696. Then, an enrichment analysis of the DEGs was conducted to reveal the potential molecular mechanisms associated with neurological outcomes after CA, and WGCNA analysis was applied to explore the relationship between DEGs and clinical parameters. The significant gene modules associated with a high CPC score were also selected. Next, PPI network analysis was performed by Cytoscape’s cytoHubba plug-in to screen out eight hub genes, and co-expressed genes and potential pathways of the eight hub genes were also investigated. These central genes were further experimentally validated by animal models and correlation analysis was performed to analyze the correlation between hub genes and NDS scores in rats. Our work may provide new and in-depth insights for predicting neurological prognosis and exploring potential molecular mechanisms.

## 2. Materials and Methods

### 2.1. Datasets Collection

The mRNA expression profiles and clinical information of GSE29540 and GSE92696 were acquired from the GEO Datasets. GSE29540 was based on the GPL13218 platform with 84 blood samples with low CPC scores (CPC 1–2) and 56 blood samples with high CPC scores (CPC 3–5). GSE92696 was also based on the platform GPL10558 with 10 blood samples with low CPC scores (CPC 1–2) and 12 blood samples with high CPC scores (CPC 3–5). CPC scores 3–5 was defined as poor prognosis of neurological function, while CPC score 1–2 was defined as good neurological prognosis function.

### 2.2. Data Normalization and Differential Expression Analysis

Before performing differential expression analysis, the data were homogenized using the R (version 4.1.1) package “preprocessCore”. After pre-processing, DEGs were screened using the “limma” package with a threshold of *p*-value < 0.05. Genes that were simultaneously upregulated or downregulated in both two datasets were selected as DEGs for subsequent analysis. The R package “ggplot2” and “pheatmap” were used for volcano plots and heatmaps respectively.

### 2.3. Functional Enrichment Analysis

Functional enrichment analysis of DEGs was carried out using the R package “clusterProfiler”, including Gene Ontology (GO) terms and the KEGG pathways. Gene set enrichment analysis (GSEA) was performed using the R package “clusterProfiler” to explore the function of the hub genes.

### 2.4. WGCNA Analysis

The co-expression network of DEGs was investigated using the R package “WGCNA”, and the gene modules associated with high CPC scores were screened using GSE92696. The gene modules most relevant to high CPC scores were selected for further PPI construction.

### 2.5. Protein–Protein Interaction (PPI) Analysis and Hub Genes Recognition

Key genes from WGCNA were inputted into the STRING (https://string-db.org/, accessed on 10 April 2022) database to build a protein–protein interaction (PPI) network. Cytoscape (V3.7.2) was used to visualize the PPI networks. The cytoHubba plugin in Cytoscape was used to identify hub genes. Three algorithms, including Degree, Stress, and Radiality were used to identify hub genes in the PPI network. The top 12 ranked hub genes of the three algorithms were selected and intersected. 

### 2.6. Immune Cell Infiltration

The ssGSEA function of the R package “GSVA” was used to evaluate the infiltration level of 23 immune cells and to explore the correlation between immune cell infiltration and each hub gene.

### 2.7. Construction of the Animal Model

A total of 21 male adult Wistar rats (350–400 g) were obtained from the Animal Experimental Center of Southern Medical University and randomly divided into sham-operated group (*n* = 7) and model group (*n* = 14). The 10-min asphyxial cardiac arrest (CA)model was performed in our study [20,21]. The rats were anesthetized with 3% pentobarbital (40 mg/kg) and intubated. The right femoral vein and femoral artery were punctured and catheterized respectively. The femoral vein was used for drug injection, and the femoral artery was used for blood pressure monitoring. After electrical cardiac monitoring was connected, the rats were given an intravenous injection of vecuronium (3 mg/kg). CA was defined as the average arterial pressure <30 mmHg. After 10 min, the rats were given extracardiac compression (200 times/min), at a depth of 1/3 of the anteroposterior diameter of the thorax and connected to ventilator-assisted ventilation (tidal volume 5 mL, 80 breaths/min). Intravenous adrenaline 0.02 mg/kg was administered every 3 min until the circulation of rats was restabilized, and ROSC was defined as the rise in mean arterial pressure of more than 60 mmHg for more than 10 min. After the rats resumed spontaneous breathing, the ventilator was removed. In the Sham group, the rats were treated with pentobarbital (40 mg/kg) and intubated, then both the femoral vein and femoral artery were catheterized. After 24 h, the neurological function of the rats was evaluated according to the international NDS score [22]. The rats were divided into NDS high group (NDS > 60) group and NDS low group (NDS < 60). The NDS high group indicated a good outcome while the NDS low indicated a poor outcome, which has been widely used in previous studies [23]. After 24 h of cardiopulmonary resuscitation, the rats were sacrificed, and subsequent experiments were performed.

### 2.8. TUNEL Staining

Three brains from each group of rats were fixed in formaldehyde (10%) and paraffin-embedded for TUNEL staining, and the rats were euthanized 24 h after the operation. Then the hippocampal region of each brain (3.2 to 4.2 mm from the bregma) was cut into sections of 10 μm sections, and two sections from each sample were randomly selected for in situ cell death detection using a Roche In Situ Cell Death Detection Kit (Roche, Penzberg, Germany) according to the manufacturer’s protocol for paraffin-embedded tissues.

Paraffin-embedded hippocampal sections were heated at +60 °C followed by washing in xylene and rehydrating through a graded series of ethanol and double-distilled water. Then the sections were placed in a plastic jar containing 200 mL 0.1 M citrate buffer and 300 W microwave irradiation was applied for 5 min. After rinsing twice with PBS, the sections were incubated with the TUNEL reaction mixture in a dark humidified chamber at 37 °C for 1 h. Then, the sections were washed with PBS and incubated for 30 min. After washing with PBS, the sections were stained with 4′-6-diamidino-2-phenylindole (DAPI) for 15 min. The number of neurons in the pyramidal layer (200 μm in length) of the CA1 region of the Hippocampal was analyzed using a fluorescence microscope equipped with the Image-pro Plus System (Media Cybernetics, Rockville, MD, USA). The number of apoptotic neurons induced by ischemia–reperfusion was determined by comparing the number of TUNEL-positive cells with the number of DAPI-positive cells.

### 2.9. Hematoxylin–Eosin (H&E) Staining and Nissl Staining

Rats were euthanized within 24 h after the cardiopulmonary resuscitation, and three rats in each group were assigned to H&E staining and Nissl staining to determine morphological changes in the CA1 region of the hippocampus. 

Brain tissue was routinely embedded in paraffin and cut into 10-μm coronal sections and brain tissue sections were stained with H&E or Nissls for routine histological examinations and the morphological changes were observed under a light microscope (IX81; Olympus, Tokyo, Japan).

### 2.10. Quantitative Real-Time Reverse Transcription PCR (qRT-PCR)

The total RNA of rat hippocampal tissue was extracted with a trizol reagent, and the RNA concentration was measured using a nanodrop 2000 instrument, followed by conversion of RNA to cDNA using Takara RT Kit (RR047a). TB Green™ Premix Ex Taq™ II(Tli RNaseH Plus) (RR820Q) was performed on the CFX96 Real-Time PCR Detection System (Applied Biosystems, Waltham, MA, USA) to evaluate relative mRNA levels. The 2-ΔΔCt method was employed to measure the relative mRNA expression (normalized to GAPDH), and all experiments were carried out three times. The primers used in this study were produced by Servicebio (Wuhan, China), and the sequences are shown in Table 1.

### 2.11. Western Blotting Assay

Four animals per group were used to assess the protein levels of RPL26, PSMD14, HINT1, RBX1, EEF1B2, RPL27, SNRPD2, and PFDN5. The hippocampus of the rat was rapidly removed and frozen at −80 °C after decapitation. Samples were homogenized in a standard lysis buffer (100 mM Tris, pH 7.4, 150 mM NaCl, 1 mM EGTA, 1 mM EDTA, 1% Triton X-100, and sodium deoxycholate 0.5%) and protease inhibitor solution (complete; Sigma-Aldrich, St. Louis, MI, USA, 05056489001) and centrifuged at 12,000 rpm for 20 min. The lowry method was selected to determine the protein concentration. The proteins (20 μg) were separated by SDS-polyacrylamide gel electrophoresis and then transferred to PVDF membranes. The membranes were blocked with 5% BSA in tris-buffered saline and then incubated at 4 °C overnight with respective primary antibodies (the primary antibodies in our studies were shown in Table 2). After washing with Tris-buffered saline, tween 20 (TBST), the membranes were incubated for 1 h with biotinylated goat anti-rabbit IgG (1:1000, Affinity; S0001) as a secondary antibody. After three washes (PBS-Tween-20, 0.05%), the membranes were incubated with the ABC Elite kit (PK6100; Vector Laboratories) for 1 h, followed by development with diaminobenzidine (D5905; Sigma). Protein expression was assessed using the freely available ImageJ software (Wayne Rasband, National Institutes of Health, Bethesda, MD, USA, version 1.51). The data obtained were normalized and reported as a percentage of normalized area relative to the internal control (β-actin) and presented as the mean of at least six independent experiments.

### 2.12. Statistics

Data were collected and analyzed. The data were statistically analyzed using GraphPad 8.0 and SPSS 26.0 software. The measurement data were expressed using the mean ± standard (M ± SD) deviation and the count data were expressed using M (Q1, Q3). In comparison, a student’s T-test was used between two groups. For comparisons between three groups, one-way ANOVA was used. While, for comparisons between the 3 groups, an analysis of variance was used. In addition, a Pearson correlation analysis was used to analyze the correlation between gene expression levels and protein expression levels of the 8 hub genes and NDS scores in rats. R^2^ > 0.9 was considered to be correlated. *p* < 0.05 was considered to be statistically different.

## 3. Results

### 3.1. Identification of DEGs

Before performing differential expression analysis, the mRNA expression data was homogenized to get more accurate results (Appendix A). The differential expression analysis identified 3550 upregulated DEGs and 3336 downregulated DEGs in GSE29540 (Figure 1A), and 1382 upregulated DEGs and 848 downregulated DEGs in GSE92696 (Figure 1B). DEGs that were simultaneously upregulated or downregulated in both datasets were selected as DEGs for subsequent analysis, with 116 upregulated and 97 downregulated genes (Figure 1C,D).

### 3.2. Functional Enrichment Analysis

Functional enrichment of these DEGs was performed to identify possible biological functions. For the results of KEGG, pathways of neurodegeneration-multiple diseases, Parkinson’s disease, Prion’s disease, and Alzheimer’s disease pathways were enriched (Figure 1E). GO analysis showed that DEGs were mainly related to positive regulation of cellular catabolic process and cell cycle checkpoints in the biological process (BP) term, with mitochondrial inner membrane and mitochondria-containing protein complex in cellular component (CC) term, and with ubiquitin-protein ligase binding and structural constituent of the ribosome in molecular function (MF) terms (Appendix A).

### 3.3. WGCNA and Identification of Key Modules

We further conducted WGCNA in GSE92696 and used DEGs to screen for key gene modules associated with high CPC scores. Figure 2A displayed the distribution of samples in GSE92696. Six were considered the best soft thresholds (Figure 2B). Similar modules were merged (Figure 2C). A total of 175 DEGs in MEgrey and Meblue were selected due to their positive correlation with high CPC scores (Figure 2D). The co-expression of DEGs in various modules was displayed in Figure 3E. GO and KEGG analyzes were further performed. The results indicated that these 175 DEGs were enriched for negative regulation of mitotic cell cycle phase transition in BP terms, mitochondrial inner membrane in CC terms, and ubiquitin-protein ligase binding in MF terms (Appendix A). For the results of KEGG, Parkinson’s disease and pathways of neurodegeneration-multiple diseases were enriched (Appendix A). 

### 3.4. Construction of PPI Network Construction and Identification of Hub Genes

A PPI network of 175 DEGs was constructed using the STRING database (Figure 3A). Using the three algorithms (Degree, Stress, and Radiality) available in the cytoHubba, the top12 highest hub genes were selected, respectively (Figure 3B–D). Finally, eight overlapping genes were identified, including RPL27 (ribosomal protein L27), EEF1B2 (eukaryotic translation elongation factor 1 beta 2), PFDN5 (prefoldin subunit 5), RBX1 (ring-box 1, E3 ubiquitin-protein ligase), PSMD14 (26S proteasome non-ATPase subunits 14), HINT1 (histidine triad nucleotide protein 1), SNRPD2 (Small nuclear ribonucleoprotein D2 polypeptide), and RPL26 (ribosomal protein L26). According to the results of the difference analysis, the heatmap displayed the logFC value of each DEG (Figure 4A). All of these genes were down-regulated in the high CPC score group. We further explored the correlation of the 8 hub genes using GSE92696. The results revealed that these genes were positively correlated with each other (Figure 4B). We also used GSE92696 datasets to analyze the potential function of hub genes in predicting the neurological outcome after CA. The results also indicated that PSMD14 had the highest area under the curve (AUC) value (0.767) in predicting the adverse neurological outcome after CA (Figure 4C).

### 3.5. Immune Cell Infiltration

The ssGSEA function of the R package “GSVA” was used to evaluate the infiltration level of 23 immune cells. We found that the infiltration level of activated B cells and immature B cells was higher in the high CPC score group (Figure 5A). The correlation of immune cell infiltration with each hub gene was explored (Figure 5B). For example, EEF1B2, HINT1, and SNRPD2 were positively correlated with infiltration levels of activated CD8 T cells. EEF1B2, PSMD14, RPL26, and RPL27 were positively correlated with the infiltration level of activated CD4 T cells.

### 3.6. Exploration of Hub Genes

The correlation between hub genes and all genes was analyzed in GSE92696. The heatmaps of the top 50 genes with positive correlations are displayed, respectively, in Appendix A. Based on the results of the correlation analysis, a single-gene Reactome-based GSEA analysis was performed. The top 20 results for each of the 8 hub genes were presented, respectively (Figure 6). For example, we predicted that PSMD14 was closely associated with eukaryotic translation elongation, RNA metabolism, cellular response to stress, and cellular response to external stimuli.

### 3.7. Validation of Hub Genes

We further constructed a rat CA model. The neurological function of rats after CA was evaluated according to the international NDS score. The rats were divided into the sham group, the NDS high group, (NDS > 60) and the NDS low group (NDS < 60). The representative pictures of H&E staining, Nissl staining, and Tunel staining were displayed in Figure 7A. The percentage of Nissl-positive cells was lower in the low NDS group than in the high NDS group and the control group, while the percentage of Tunel-positive cells was highest in the low NDS group (Figure 7B). We next examined the mRNA and protein levels of the eight hub genes in the three groups. The results indicated that the mRNA and protein levels of the eight hub genes were all lowest in the low NDS group and highest in the control group, which was consistent with our analysis results (Figure 7C,D). 

### 3.8. Correlation Analysis

Furthermore, the correlation between gene expression levels and protein expression levels of the eight hub genes and NDS scores in rats was also analyzed. Our results showed a positive correlation between the protein expression of PSMD14, RPL26, PFDN5, RBX1, SNRPD2, and the NDS scores of rats, whereas there was no correlation between the protein expression levels of EEF1B2, RPL27, HINT1, and the NDS score of rats (Figure 8A). In addition, the mRNA levels of all these eight genes were significantly and positively correlated with the NDS score (Figure 8B).

## 4. Discussion

Cardiac arrest is common, and the consequences of it can be devastating. The cause of cardiac arrest is most often cardiogenic disease, the others including respiratory insufficiency, neurological causes, and so on [24,25]. Hypoxic–ischemic brain injury is a leading cause of mortality and disability in comatose patients after cardiac arrest. Although the current study showed that the ideal therapy and neuroprotection of temperature management are uncertain, there was still no sufficient evidence to recommend for or against temperature control at an appropriate temperature or early cooling after cardiac arrest, more work is needed [26,27]. As the management of a coma after cardiac arrest has improved during the past decade, an increasing proportion of patients are surviving than ever before. Neurological prognostication remains a significant clinical challenge in postresuscitation care. Serum biomarkers released by brain cells after hypoxic–ischemic injury may help to predict prognosis [28]. Presently, the biomarker recommended in the prognosis guidelines of the European Resuscitation Council prognostication guidelines is neu-ron-specific enolase (NSE), but NSE has limitations [29], and there is a lack of other validated predictive biomarkers. Additionally, relying on a single indicator is unreliable in achieving better prognostication. Multimodal assessment based on the integration ofintegrating of different indicators is recommended to increase the predictive value in identifying patients with unfavorable neurological outcomes [30,31]. Identifying novel biomarkers may provide important insights into the evaluation of prognosis. Therefore, in this study, we aimed to screen new biomarkers and potential molecular mechanisms affecting the prognosis of early neurological function after CA. 

Recently, research showed that transcriptomics plays an essential role in assessing neurological prognosis with brain injury. Meng et al. found that genomic information has the potential in predicting trauma brain injury (TBI) pathogenesis, and hippocampal transcriptome signatures were assosiated with behavior phenotypes [27]. Osier et al. explored the relationship between biomarker-encoding genes and TBI outcomes, and their study showed that variation in candidate biomarker genes is associated with neurological outcomes after brain injury [32]. In this work, we used a large-scale genome-wide expression analysis to determine whether genetic features are associated with outcomes in patients with CA. Gene expression data and clinical information for GSE29540 and GSE92696 were downloaded from the GEO database and used to explore critical genes and pathways associated with poor outcomes in CA patients. Finally, a total of 116 upregulated and 97 downregulated DEGs were selected. These DEGs were enriched in BP terms for cytolytic processes and cell cycle checkpoints, in CC terms for mitochondrial inner membranes and mitochondria-containing protein complexes, in MF terms for ubiquitin–protein ligase binding and structural components of ribosomes, and in KEGG pathways for neurodegeneration-multiple disease, Parkinson’s disease, Prandial disease, and Alzheimer’s disease pathways. Further, the WGCNA screening of GSE92696 identified 175 DEGs associated with high CPC scores. GO enrichment and KEGG pathway enrichment analyses of these 175 DEGs were also consistent with these results. RNA-seq has been widely used to screen for differential genes in brain ischemia–reperfusion injury and to help unravel the molecular mechanisms involved. For example, Dergunova et al. used high-throughput RNA-seq to analyze differential genes in brain tissue from a transient MCAO model and, finally, identified 469 genes with greater than 1.5-fold differences 4.5 h after trauma and 1469 genes with greater than 1.5-fold differences 24 h after trauma [33]. In addition, Lyu et al. used RNA-Seq to reveal the specific molecular mechanisms of ischemia–reperfusion injury in the brain treated with Semax peptide and Shuxitong injection. However, unlike these RNA-Seq studies using tissue RNA, the two GSE profiles we analyzed were aimed to analyze blood gene expression in CA patients, and the blood gene expression profile facilitates the discovery of new serological gene markers and is used for prognostic prediction of this disease [34,35].

To date, our study is the first work performed to analyze the gene expression profiles of GSE29540 and GSE92696 and reveal important prognostic markers. The PPI network of 175 DEGs was constructed using the STRING database using three algorithms (Degree, Stress, and Radiality). Finally, eight overlapping genes were identified, including RPL27, EEF1B2, PFDN5, RBX1, PSMD14, HINT1, SNRPD2, and RPL26, which were downregulated in the high CPC score group. To the best of our knowledge, all these genes have not ever been studied in CA. We further explored the correlation of the eight hub genes and revealed that these genes were positively correlated with each other. Then, we analyzed the potential function of hub genes to predict the outcome of patients with CA. EEF1B2, PSMD14, HINT1, and SNRPD2 were proved to be eligible predictors (AUC > 0.7). It was reported that a novel biallelic loss of EEF1B2 function was reported to be associated with autosomal recessive intellectual disability [36]. In addition, EEF1B2 was also a potential biomarker for the pathogenesis of Alzheimer’s disease [37]. Among these hub genes, PSMD14 had the highest AUC value (0.767). Studies on PSMD14 were mainly associated with tumors, for example, targeting PSMD14 inhibited melanoma growth through SMAD3 stabilization [38]. Previously, Wang et al. used miRNA-seq to identify miRNAs in the sera of stroke patients that could influence patient prognosis, and the results of this study showed that miR-328-3p levels were an independent correlate of short-term prognosis [39]. Generally, the close association of these genes with the nervous system suggests that the relationship between the expression status of these genes and the prognosis of the disease is worth investigating. Their detailed molecular mechanisms are still needed to reveal further the specific roles of the eight hub genes in this disease.

In the early stages of post-cardiac arrest syndrome, the immune cell reactivity could explain the potent role of rapid hypothermia [40]. We found that the infiltration levels of activated B cells and immature B cells were higher in the high CPC score group, suggesting a potential connection between the level of B-cell infiltration and the outcome in patients with CA. Previously, the study by Kreimann et al. showed that ischemia/reperfusion injury after allogeneic renal transplantation triggers CXCL13 release and B-cell recruitment [41]. In addition, the study by Renner et al. showed that B-cell subsets contribute to renal injury and renal protection after ischemia/reperfusion. Thus, immune cell infiltration and inflammatory responses may play an important role in ischemia–reperfusion injury of the brain, and the specific functions and detailed molecular mechanisms of these immune cells will be further revealed in the future.

We further constructed a rat CA model to validate the expression of the eight hub genes. The neurological function of rats after CA was evaluated according to the international NDS score. We examined the mRNA and protein levels of the eight hub genes and found that the mRNA and protein levels of the eight hub genes were all lowest in the low NDS group and highest in the control group, which was consistent with our analysis results. In addition, we performed a Pearson correlation analysis to investigate the correlation of gene expression profiles with NDS scores and prognosis. The results showed a positive correlation between the protein expression of PSMD14, RPL26, PFDN5, RBX1, SNRPD2 and the NDS score of rats, whereas there was no correlation between the protein expression level of EEF1B2,HINT1,RPL27 and the NDS score of rats. In addition, the mRNA levels of all these eight genes were significantly and positively correlated with the NDS score. This is the first study is to correlate gene expression profiles with NDS scores in this field. Previously, Osier et al. reported the associations between genetic polymorphisms related to blood-based biomarkers and neurological outcomes after TBI; in their study, the variation in S100B was found to significantly predict the outcome [32]. However, the relationship between biomarker-encoding genes and cardiacl arrest outcomes remains unknown. Our study shows that epigenomic programming and transcriptional activities in the brain were perturbed after cardiac arrest, triggering the brain injury by a gene-dependent mechanism. Eight genes were identified, and high expressions of them in the brain or blood are associated with a good neurological outcome after cardiac arrest, providing novel insight regarding mRNA involvement in biomarkers for predicting a neurological prognosis following cardiac arrest.

There are limitations to this study, and more future research is needed. The ideal biomarker should be easy to detect, stable, and sensitive. Biomarkers in peripheral blood are often available [42]. In the experimental validation, we only detected the mRNA level and protein level in the hippocampus of rats. Whether these genes’ mRNA expression level and protein level are consistent with the prognosis in the blood of patients with cardiac arrest still needs to be studied in future clinical experiments. The expression of these genes was detected at one time point. Perhaps a continuous monitoring of them can better clarify their potential effects and mechanism.

## 5. Conclusions

In summary, our study identified eight potential biomarkers predicting neurological outcomes after cardiac arrest and investigated their potential pathways and the correlation of these gene expression profiles with NDS scores. Our work provides a new direction for future research.

## Figures and Tables

**Figure 1 brainsci-12-00928-f001:**
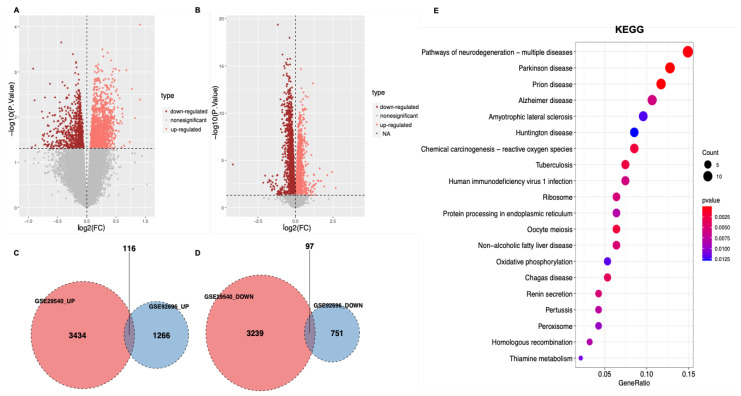
Identification of DEGs and enrichment analysis of the DEGs. (**A**) The volcano plot of GSE29540. The orange points show upregulated genes, and the coffee-colored points represent downregulated genes. (**B**) The volcano plot of GSE92696. (**C**,**D**) Venn diagrams of the number of upregulated (**C**) or downregulated (**D**) DEGs in both datasets. (**E**) KEGG analysis of overlapping DEGs. The top 20 terms are displayed. *p*-value < 0.05 was set as the critical value.

**Figure 2 brainsci-12-00928-f002:**
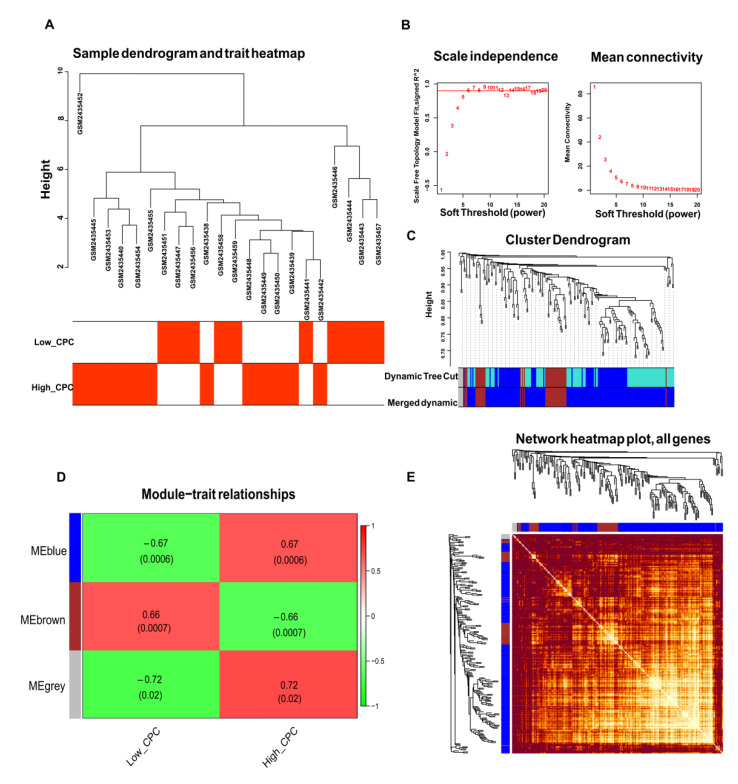
Weighted gene co-expression network analysis (WGCNA) and identification of the key modules via WGCNA. (**A**) The distribution of samples in GSE92696. (**B**) The screen of the best soft thresholds. Six was considered the best soft threshold. (**C**) The merging of similar modules. (**D**) Heatmap of the correlation between the module and the clinical characteristics of the patients in GSE92696. (**E**) The co-expression of DEGs in each module.

**Figure 3 brainsci-12-00928-f003:**
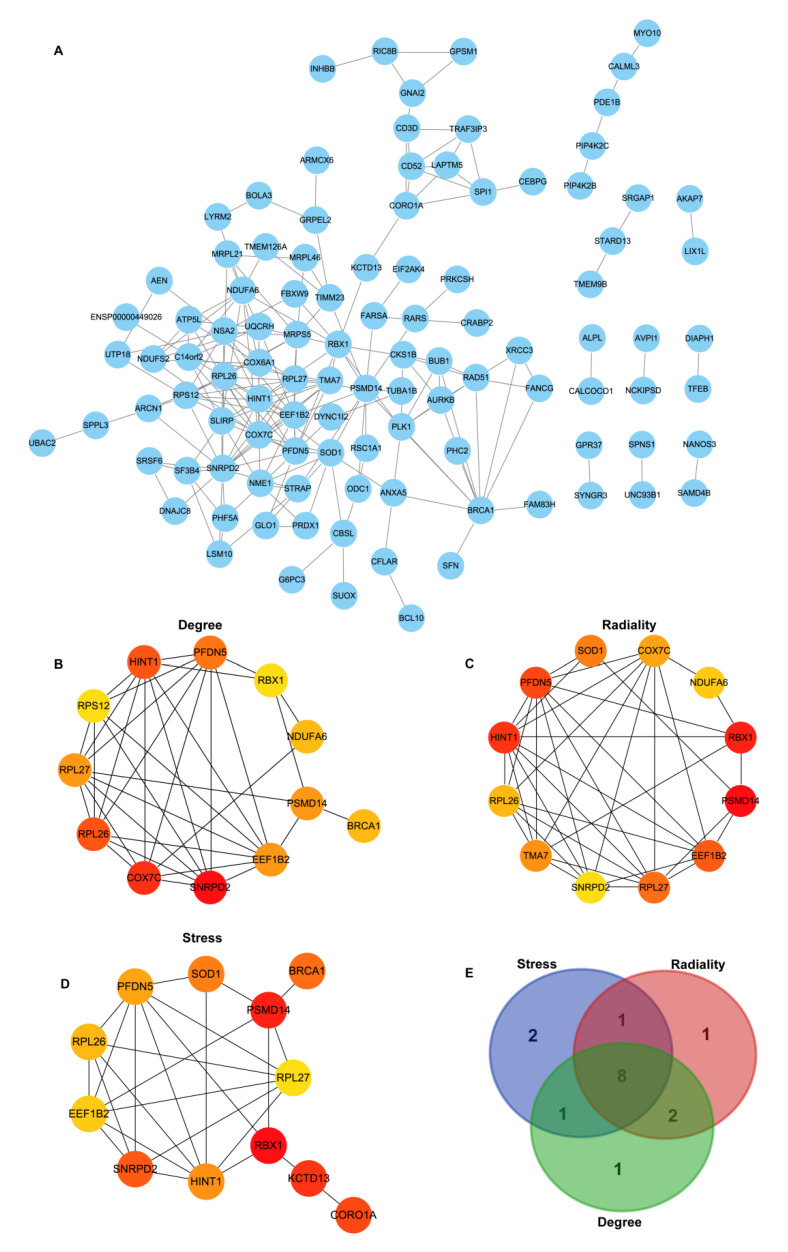
PPI network for DEGs and the identification of hub genes. (**A**) The PPI network was constructed from DEGs using the STRING database. (**B**,**C**) Top 12 hub genes identified by different algorithms in cytoHubba, including Degree (**B**), Radiality (**C**), and Stress (**D**). (**E**) Common hub genes identified between the 3 algorithms.

**Figure 4 brainsci-12-00928-f004:**
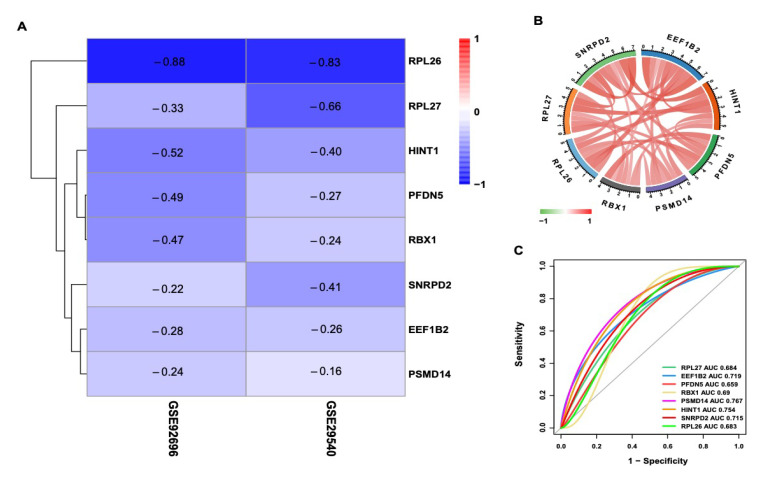
Validation of the hub genes. (**A**) Heatmap displays the log2(FC) values of the hub genes in GSE92696 and GSE29540. (**B**) Correlation of 8 hub genes in GSE92696. The red line represents a positive correlation, the green represents a negative correlation, and the deeper the color, the stronger the correlation. (**C**) The ROC curves of the hub genes in GSE92696.

**Figure 5 brainsci-12-00928-f005:**
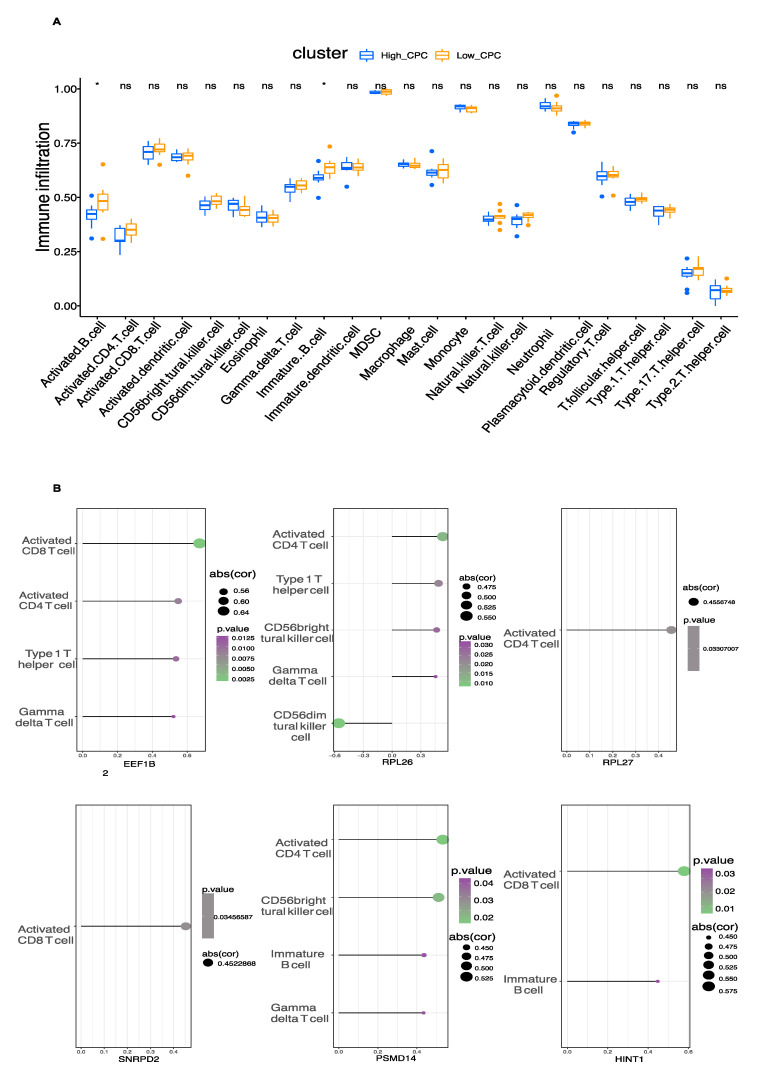
Immune cell infiltration. (**A**) The differences in the level of infiltration of 23 immune cells between the high and low CPC score groups. (**B**) The correlation of the hub genes with the level of infiltration of 23 immune cells was indicated. * *p* < 0.05.

**Figure 6 brainsci-12-00928-f006:**
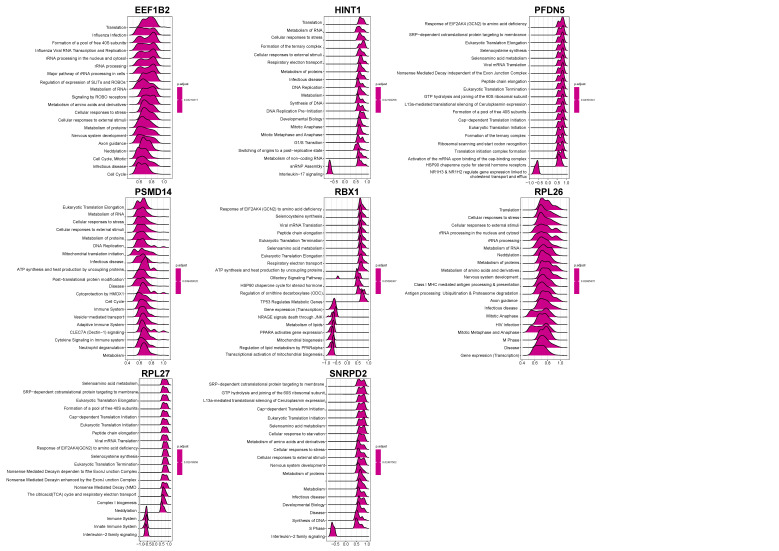
GSEA of the hub genes. The top 20 terms of the GSEA results for the indicated hub genes are shown.

**Figure 7 brainsci-12-00928-f007:**
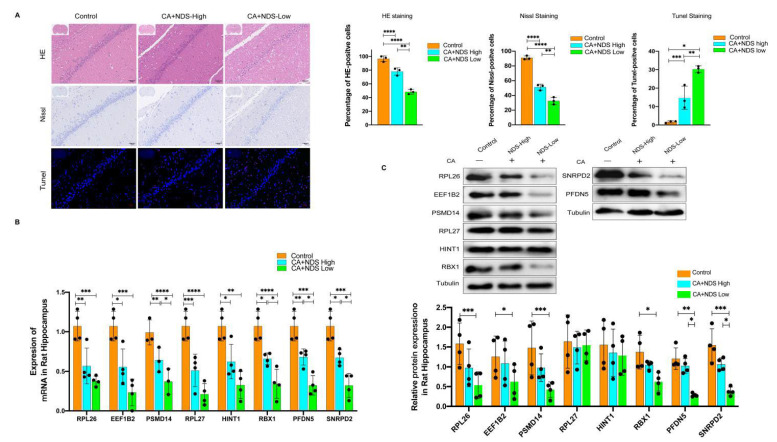
Validation of the hub genes. (**A**) Representative pictures of H&E staining, Nissl staining, and Tunel staining. (**B**) The mRNA expression of the 8 hub genes in the indicated groups. (**C**) Protein levels of the 8 hub genes in the indicated groups. Data are expressed as the mean ± SD.* *p* < 0.05, ** *p* < 0.01, *** *p* < 0.001, and **** *p* < 0.0001.

**Figure 8 brainsci-12-00928-f008:**
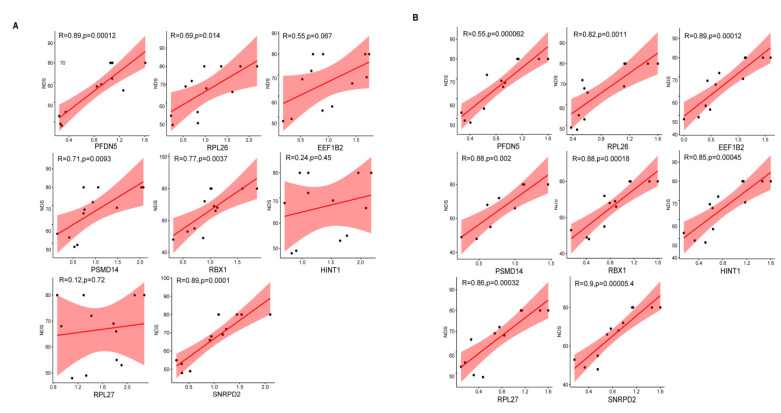
Correlation analysis between gene expression levels and protein expression levels of the 8 hub genes and NDS scores in rats. (**A**) Protein expression levels of the 8 hub genes and NDS scores in rats. (**B**) Gene mRNA expression levels of the 8 hub genes and NDS scores in rats.

**Table 1 brainsci-12-00928-t001:** Primer sequences.

Gene Name	Sequence
*RPL26*	*F: TCACCCCAGCAAGGTGGTTAT*
	*R: TGCCCTTCTCCTTTCCTACTTG*
*PSMD14*	*F: AAACAGGAAGGCCCGAGATG*
	*R: CACACCAGAAAGCCAACAGC*
*HINT1*	*F: GCAGATGATGATGATGAAAGTCTTC*
	*R: CCCGTCTGCACCTTCATTCA*
*RBX1*	*F: TTGGGGAGTCTGTAACCACG*
	*R: CTCTGTTGTCCAAGGGGCAC*
*EEF1B2*	*F: AGCTACATTGAGGGGTACGTG*
	*R: TACCAACGCAGAGCATGACA*
*RPL27*	*F: CCTCATGCCCACAAGGTACT*
	*R: AAACTTGACCTTGGCCTCCC*
*SNRPD2*	*F: CTCTCGGTGCTCACACAGTC*
	*R: ACCATGTTGCAGTGCCTGTC*
*PFDN5*	*F: AGACAGCTGAGGATGCCAAG*
	*R: TCATTTCCACGACGGCTTGT*
*β* *-actin*	*F: CACCCGCGAGTACAACCTTC*
	*R: CCCATACCCACCCATCACACC*

**Table 2 brainsci-12-00928-t002:** The information of the primary antibodies used in this study.

Name	Corporation	Co. Number	Dilution Ratio
Anti-RPL26	Proteintech	16487-1-AP	1:800
Anti-PSMD14	Abcam	Ab109130	1:1000
Anti-HINT1	Proteintech	10717-1-AP	1:500
Anti-RBX1	Abcam	Ab133565	1:1000
Anti-EEF1B2	Proteintech	10483-1-AP	1:500
Anti-RPL27	Proteintech	14980-1-AP	1:500
Anti-SNRPD2	Abcam	Ab198296	1:500
Anti-PFDN5	Proteintech	15078-1-AP	1:500
Anti-Tubulin	Beyotime	AF1216	1:1000

## Data Availability

All data included in this study are available upon request from the corresponding authors.

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
