# Peer review of "Identification and Validation of Novel Potential Pathogenesis and Biomarkers to Predict the Neurological Outcome after Cardiac Arrest"

_brainsci, 2022, doi:10.3390/brainsci12070928_

Round 1

Reviewer 1 Report

In this study Qiang et al. investigated novel potential pathogenesis and biomarkers to predict the neurological outcome after cardiac arrest. The study is well conducted and the findings are well exposed.

I have only some minor reviews to be addressed:

In the introduction section I would suggest to briefly underline the detrimental role of severe hyperoxia on neurological outcome after cardiac arrest (two recent metanalysis have been published on the topic by La Via et al. [Minerva Anestesiologica] and Holmberg et al. [Resuscitation]) and on the effect of target temperature management (again two recent meta-analysis published by Sanfilippo et al. [Journal of Clinical Medicine] and Fernando et al. [Intensive Care Medicine]).

Throughout the paper I found several syntax and punctuation mistakes that need to be corrected. For example: Lines 66, 67, 121, 122, 127, 201, 273.

Then, I have no further comments.

Author Response

Dear Reviewer,

  Thank you very much for your valuable comments on our paper.Belows are our replies to your comments.

Response 1: We really appreciate your suggestion to add the content about the role of severe hyperoxia and target temperature management on neurological outcome after cardiac arrest in the introduction section.For both shockable and nonshockable rhythms, targeted temperature management (TTM) and avoid exposure to severe hyperoxemia are generally considered necessary care after cardinal arrest due to their funciton in improving neurological outcomes.However,recent systematic reviews showed that TTM benefits on neurological outcome is uncertain,maybe further research are required to find the ideal therapy of temperature management.All these points of view above have been added into the introduction section.

Response 2:Thank you very much for your kind guidance.We have corrected these mistakes.Best Regards!

Reviewer 2 Report

Dear authors, congratulations on your work. I have some comments to improve the general readability of you paper:

-please first write in full the word and then add the acronym.

-In the description of the animal model please detail the difference between the sham group and the model group.

-There are a lot of figures in this manuscript; however they are often small and difficult to read. perhaps you could select the most representative ones and add the rest as supplemental material. 

- in the discussion, please highlight the clinical importance of you findings and add a paragraph listing the potential limitations of your study

Author Response

Dear Reviewer,

Thank you very much for your valuable comments on our paper.Belows are our replies to your comments.

Response 1:Thanks a lot for your kind help.We have used the standard abbreviations the  according to your requirements.

Response 2.The difference between the sham group and the model group is that  based on the sham operation group, asphyxia was induced by muscle relaxants and then cardiopulmonary resuscitation was performed.

Sham group:The rats were treated with 3% pentobarbital (40 mg/kg) and intubated,then both the femoral vein and femoral artery were catheterized.No muscle relaxant was used to induce asphyxia.

Model group:The rats were anesthetized with 3% pentobarbital (40 mg/kg) and intubated. The right femoral vein and femoral artery were punctured and catheterized respectively. The femoral vein was used for drug injection, and the femoral artery was used for blood pressure monitoring. Then the rats were given an intravenous injection of vecuronium(3 mg/kg).cardiacl arrest was defined as the average arterial pressure <30 mmHg.After 10 minutes , cardiopulmonary resuscitation was performed in rats.

Response 3:We greatly appreciate your suggestion.We have deleted some figures appropriately in this manuscript,and we also have improved the quality of some pictures for better reading.

Response 4:Surely,we have added the relevant contents you mentioned to discussion  section.

 Clinical importance:Osier,etal.first report the associations between genetic polymorphisms related to blood-based biomarkers and neurological outcome after trauma brain injury,in their study variation in S100B was found to significantly predict outcome.However,the relationship between biomarker-encoding genes and cardiacl arrest outcomes remains unknown. Our study shows that epigenomic programming and transcriptional activities in brain were perturbed  after cardical arrest,which triggers the brain injury by a gene-dependent mechanism. Eight genes were identified and high expressions of them in  brain or blood are associated with a good  neurological outcome after cardical arrest,These results provide novel insight regarding mRNA' involvement in biomarkers for prediction of neurological prognosis following cardiac arrest.

 The limitations of this study:he ideal biomarker should be easy to detect, stable and sensitive,biomarkers in peripheral blood are often available. In the experimental validation, we only detected the mRNA level and protein level in hippocampus of rats,whether the mRNA expression level and protein level of these genes are consistent with the prognosis in blood of patients with cardiac arrest still needs to be studied in future clinical experiments. The expression of these genes was detected at one time point,perhaps a continuous monitoring of them can better clarify their potential effects and mechanism.

Reviewer 3 Report

In the manuscript, the authors aimed to identify novel biomarkers “capable” of predicting neurological prognosis after CA.

The paper is very interesting, and I would like to suggest few modifications before being published. Comments:

(1)   The use of biomarkers in CA prognosis is very interesting, well integrated in a multimodal approach for neurological prognostication. As stated in the guidelines and as reported in this paper, NSE is the only biomarker routinely used in clinical practice; are there any differences with the biomarkers investigated in this study? According to the author, these biomarkers could express different pathogenic pathways compared to NSE? If so, why potentially adding these biomarkers to the clinical practice may improve the prognostication? Not always adding parameters will give us real new information. Maybe a couple of lines in the discussion will help to give a clinical imprint of the very nice results.

(2)   Low-quality images, is it possible to improve them? Maybe not all the figures are essential

(3)   Add reference for the worldwide incidence of CA (line 34)

(4)   Minor grammar and layout issues:

-        line 42 repetition of clinical and clinically

-        line 46 “has come to the forefront” I suggest rephrasing it

-        line 121 “as described previously”: I cannot find the previous explanation

-        line 350 “Further” with a capital letter

Author Response

Response 1.Thank you for your nice comments on our article.According to the comments, we have made extensive modifications to our manuscript.The main points are as follows:

Although NSE is the most widely studied measure for predicting neurological outcomes in clinical practice,but the guide  did not recommend the detection method, inconsistent threshold and determined detection time.In addition,NSE is also expressed in red blood, neuroendocrine and small-cell lung cancer cells, increasing the risk of falsely increased serum levels in cases of haemolysis or in the presence of neuroendocrine tumours,Moreover, NSE shows an acceptable, but still only moderate, sensitivity at high specificity and performs poorly before 48 h after cardiac arrest.Relying on single indicator is unreliable in achieving better prognostication.Multimodal assessment based on the integration of different indicators is recommend to increase the predictive value in identifing patients with unfavorable neurological outcome,identifying novel biomarkers may provide Important Insights into the evaluation of prognosis.Recently,researches showed that transcriptomics plays a important role in the assessment of neurological prognosis with brain injury and that variation in candidate  biomarker genes are associated with neurological outcomes after brain Injury. Gene expression profiling has the capability to identify biomarkers associated with response to brain injury after cardical arrest.Hence,we focused on finding gene-biomarker that  has the potential to predict the prognosis of neurological function.These findings represent a novel contribution to the evidence that can inform future studies aimed at enhancing interpretation of biomarker data, identifying novel biomarkers, and ultimately harnessing this information to improve clinical outcomes and personalize care.

Response 2.We greatly appreciate your suggestion.We have deleted some figures appropriately in this manuscript,and we also have taken certain measures to improve the quality of images.

Response 3.We have add reference for the worldwide incidence of CA.(line 35)

Response 4.We feel sorry for our carelessness. In our resubmitted manuscript, the error is revised. Thanks for your nice suggestions.

 The mistake “repetition of clinical and clinically” has been corrected (line 51).

The words “has come to the forefront” we have rephrased as “neuron-specific enolase (NSE) is the most widely studied measure for predicting neurological outcomes after CA" (line 56,line 67).

The points we want to express on this is that the animal model method in our study was consistent with the previous studies, we have listed relevant references ( line 135).

“further ” has been replaced with“Further”  (line 372).

Reviewer 4 Report

The paper presented by the authors concerns an important issue - estimating the risk of complications and the chance of returning to normal functioning in patients after cardiac arrest. Both the idea of ​​the manuscript and its implementation are correct. Reading the papers, I got the impression that the implementation is at least difficult for several reasons which, in my opinion, should be improved.
- I did not find information on the etiology of sudden cardiac arrest in the manuscript.
- lack of information on the undertaken therapeutic activities
- there is no information on the effect of mild therapeutic hypothermia.
I believe this information must find a place in the manuscripts. Cardiac arrest in the hospital has a different prognosis than outside, in the course of coronary heart disease and rhythm disturbances (of what types?). A recognized method of treatment is mild therapeutic hypothermia used in selected patients.
The above elements must be included in the publication in order to constitute a multidisciplinary link from bench to bedside.

Author Response

Dear reviewer,

We feel great thanks for your professional review work on our article.

As you are concerned, there are several problems that need to be addressed.

the detailed corrections are listed below.

Point 1. Add the content about the information on the etiology of sudden cardiac arrest in the manuscript.

Response 1:Thank you for your nice comments on our article.According to the comments, we have added the suggested content to the manuscript.In this study, we focused on brain injury after cardiac arrest,but It is still necessary to refer to the cause of cardiac arrest to make this article more standardized.Cardiogenic diseases and respiratory diseases are the common causes of all cardiac arrest.The modified information is as follows

Line 341-343. “Cardiac arrest is common and the consequences of it can be devastating. The cause of cardiac arrest is most often cardiogenic disease,others including respiratory insufficiency,neurological causes and so on”

Point 2.Lack of information on the undertaken therapeutic activities.

Response 2.According to your suggestion, we added the content about therapeutic activities after cardiac arrest in the introduction section, such as better emergency care,timely transport,avoiding exposure to severe hyperoxemia,implementing of targeted temperature management.As we mainly focus on the neurological prognosis after cardiac arrest In this article,we also emphasized the importance of predicting neurological outcomes with the backgroud that more patients survived from cardiac arrest.the detailed corrections in the draft are listed below.

Line 36-42. “More people are surviving cardiac arrest,and the overall results have improved through better emergency care, including early cardiopulmonary resuscitation (CPR), timely transport, and the implementation of a post-resuscitation care bundle [2,3]. Post-cardiac arrest brain injury is still a substantial cause of morbidity and mortality. For both shockable and nonshockable rhythms, targeted temperature management (TTM) and avoiding exposure to severe hyperoxemia are generally considered standard care after CA because they improve neurological outcomes.”

Point 3. There is no information on the effect of mild therapeutic hypothermia.
Response 3.

 As we all know,the pecognized method of treatment is mild therapeutic hypothermia used in selected patients atfer cardiac arrest,but some recent systematic review showed that TTM at 32-34°C does not improve survival nor neurological outcome after CA and increases the risk of arrhythmias. Now, there was insufficient evidence to recommend for or against temperature control at 32-36°C or early cooling after cardiac arrest. In addition,the optimal therapeutic duration of targeted temperature management remains unclear. The dose of TTM is also unclear raising the on-going question about the dose of targeted temperature management. ERC-ESICM (European Resuscitation Council and the European Society of Intensive Care Medicine) guidelines on temperature control after cardiac arrest in adults recommend not actively rewarming comatose patients with mild hypothermia after return of spontaneous circulation (ROSC) to achieve normothermia.More work is needed in future.According to your suggestion,we have revised the manuscript extensively,the detailed information is below.

Line 40-44. “For both shockable and nonshockable rhythms, targeted temperature management (TTM) and avoiding exposure to severe hyperoxemia are generally considered standard care after CA because they improve neurological outcomes.However, recent systematic reviews showed that TTM benefits on the neurological outcome are uncertain.”

Line 344-345. “Although the current study showed that the ideal therapy and neuroprotection of temperature management are uncertain, there was still no sufficient evidence to recommend for or against temperature control at appropriate temperature or early cooling after cardiac arrest.” 
